# Adherence to Personalised Nutrition Education Based on Glycemic and Food Insulin Index Principles and Their Association with Blood Glucose Control in Individuals with Type 2 Diabetes Mellitus

**DOI:** 10.3390/ijerph22060925

**Published:** 2025-06-11

**Authors:** Hildegard Strydom, Jane Muchiri, Elizabeth Delport, Zelda White

**Affiliations:** 1Department of Human Nutrition, University of Pretoria, Pretoria 0084, South Africa; jane.muchiri@up.ac.za (J.M.); zelda.white@up.ac.za (Z.W.); 2GI Foundation of South Africa, Mbombela 1201, South Africa; liesbet@gifoundation.com

**Keywords:** personalised nutrition education, glycemic index, glycemic load, food insulin index, type 2 diabetes, blood glucose control

## Abstract

Personalised nutrition education (PNE) can enhance blood glucose control (BGC). We determined whether patients with type 2 diabetes (T2DM) adhered to PNE based on glycemic index (GI), glycemic load (GL), and food insulin index (FII) principles and whether adherence was associated with improved BGC. This retrospective cohort included 67 files for patients who received PNE. The patients completed 3-day food and blood glucose records at three points over 90 days. HbA1c values were compared between time points. An adherence score sheet (ASS) was used to determine their adherence to PNE and the main meal adherence classification (MMAC). A one-way repeated measures ANOVA was used to assess the changes over time. A chi-square test determined the association between the MMAC and blood glucose levels falling within the targeted ranges. Correlations between dietary adherence and BGC indicators were examined using Pearson’s product–moment correlation. Adherence ranged from 88 to 95%. MMAC score was significantly associated with blood glucose being within the targeted ranges (*p* = 0.028). Mean blood glucose decreased over time, but the correlations with adherence were only significant at time point 1 (*p* = 0.029). HbA1c levels decreased significantly over time (*p* = 0.003), but their correlation with adherence was not significant (*p* > 0.05). In patients with T2DM, high adherence to PNE based on GI, GL, and FII principles was associated with improved BGC.

## 1. Introduction

Continuous rises in T2DM to epidemic proportions [1,2,3] and the devastating effects of diabetic complications like cardiovascular disease, neuropathy, kidney failure, and diabetic retinopathy [4] highlight the need for effective dietary interventions to manage T2DM.

Diet therapy is a critical aspect of the treatment and self-management of T2DM and aims to follow the best-evidence-based approach in terms of nutrition education for diabetic patients. In a recent systematic review and meta-analysis on the effects of dietary education interventions on type 2 diabetics, it was found that personalised dietary education (including contact and non-contact education) was the most effective of all dietary interventions for the dietary treatment of T2DM [5]. In addition, interventions provided for at least 3 months effectively controlled HbA1c levels [5]. Therefore, nutrition education must not only be personalised but must also be provided over longer periods to improve the blood glucose control in patients with T2DM. In addition, PNE is also preferred over group education sessions [6].

Adherence to diabetes nutrition education has been shown to lower HbA1c, blood pressure, and body mass index (BMI) values [7,8]. Carbohydrates are currently the dietary macronutrient considered to have the most profound effect on postprandial blood glucose and insulin levels [9,10,11,12,13]. Studies on the digestion of carbohydrates have clearly illustrated that not all carbohydrates have the ability to increase plasma glucose levels to the same extent (and therefore cause hyperinsulinemia equally). The importance of using the GI to predict the postprandial glucose and insulin effect of single-carbohydrate-containing foods on blood glucose control has been demonstrated [11,14,15]. Equally, the GL, which takes into account the GI and the grams of carbohydrates consumed in a meal or snack or per day, has proven to be one of the best predictors of blood glucose responses to meals [9,16]. The beneficial effects of low-GI diets for decreasing postprandial blood glucose and insulin levels in the prevention and dietary management of diabetes have also been well-described [17,18,19,20,21,22]. However, the protein and fat content of a meal can also significantly affect postprandial glucose and insulin levels [17,23,24,25,26].

Holt and colleagues confirmed in 1997 that foods containing similar amounts of carbohydrates produced very different insulin scores by introducing the food insulin index (FII) [25]. The ingestion of high-protein food items can induce insulin responses similar to those of a high-carbohydrate meal [17,25,27]. Testing of the FII found that adding fat had an inverse effect on blood glucose and insulin levels [25]. Evidence suggests that high-fat meals will reduce the release of glucose in the early postprandial state (60–90 min) and delay peak blood glucose levels to 3 h postprandially [11,26,28,29] but will then cause sustained late hyperglycaemia and insulinemia, which can contribute to the development of insulin resistance [11]. This evidence suggests that the amounts of carbohydrates, protein, and fat and their insulin secretion ability should be considered in the personalised dietary treatment of diabetes.

This study aimed to determine the dietary adherence to PNE based on GI, GL, and FII principles and whether dietary adherence was associated with improved blood glucose control in patients with T2DM.

## 2. Materials and Methods

In this retrospective cohort study, patient records for 67 adults (18–85 years) with T2DM who attended a specific dietetic private practice in Pretoria (South Africa) from January 2015 to December 2022 were randomly selected. Ethical approval was obtained from the Research Ethics Committee, the Faculty of Health Sciences, the University of Pretoria (No: 392/2023).

G*Power 3.1.9.7 was used for the sample size calculation. The conventional power level of 0.8 and a 0.05 level of significance were specified. The sample size of 67 was calculated for a medium effect of 0.3, based on Cohen’s guidelines [30] for effect sizes for correlation coefficients.

The patient population received PNE as part of their dietary consultation visits, and they took 3-day blood glucose and food records over several time points. Records were excluded if patients had disabilities preventing the normal oral ingestion of food; were pregnant or lactating; changed their exercise routines by performing double the duration of their pre-existing exercise times or more; did not complete the 3-day food and blood glucose records; or received antibiotic or cortisone treatment during the time period.

The following procedure was followed during the individualised dietetic consultation visits where PNE was given:

Time point 0 (baseline): The biometric data for each patient was recorded (age, gender, height, weight, body fat percentage, muscle mass, BMI). Height was measured using a Hi-Care wall-mounted stadiometer (KaWE Intelligent Equipment Co., Ltd., Wuxi, Jiangsu Province, China), and body weight and body composition (body fat percentage and muscle mass) were measured using an Innerscan Tanita Body Composition Monitor BC-731 (TANITA, Tokyo, Japan), which showed acceptable validity compared to DEXA (the standard for determining body composition) [31,32]. Medication, exercise regimes, and HbA1c values (when available) were recorded. Patients received a 90 min one-on-one nutrition education session presented by the dietitian/researcher. Patients who did not possess a standardised glucometer received an Accu-Check Instant glucometer (Roche Diabetes Care, Inc., Mannheim, Germany) and were trained on the correct procedure for self-monitoring blood glucose. Patients were given 30 days to implement their PNE at home while being encouraged to send questions to the dietitian to clarify any queries about the PNE. Patients were asked to complete a 3-day food and blood glucose record three days before each follow-up visit to the practice. In the 3-day records, all food and drinks consumed over 3 consecutive days and blood glucose values measured using glucometers before and 90 min after main meals had to be documented. The 3-day records were completed every 30 days (±2 weeks).

Time points 1, 2, and 3 (30, 60, and 90 days after PNE, ±2 weeks): The consultations were 30–60 min long. The measurements of height, weight, body composition, and BMI were repeated. Medication use and exercise regimes were recorded. The 3-day blood glucose records were checked to see whether the blood glucose levels fell within the targeted pre- and postprandial blood glucose level ranges [33]. All 18 blood glucose readings (this included the pre- and postprandial readings for each of the three main meals per day over three days) were used to calculate the mean blood glucose levels over the 3 days. The results were discussed with the patients, and suggestions were made where dietary changes were needed. Repeat tests of the HbA1c levels measured at 60 and 90 days (±2 weeks) were recorded if available.

### 2.1. Personalised Nutrition Education

The 90 min PNE session was provided by one of the investigators, all of whom were dietitians experienced in diabetes management. The PNE included visual aids for use at home. The content covered the causes of the development of insulin resistance and type 2 diabetes, the complications of diabetes, the basics on carbohydrates (their functions, sources, digestion, and impact on blood glucose levels), and the GI and GL concepts and how food with a lower GI and GL per serving can be used to manage blood glucose levels. Lists of food options and corresponding portion sizes were given to patients. These lists could be used as exchange lists to give patients various choices. Portion sizes were calculated to ensure that the GLs of main meals (breakfast, lunch, and supper) were no more than 25 and that the GLs of snacks were no more than 10. The portion sizes for the protein content of main meals for female patients were ≤27 g and 37 g for male patients, and the protein content of snacks was ≤7 g. The portion sizes for the fat content of main meals for female patients were ≤14 g and 16 g for male patients, and ≤3 g was allocated to snacks. The portion sizes for protein- and fat-containing foods that induced high insulin secretion during FII testing were limited.

### 2.2. Data Management

The data from each patient’s 3-day records were captured in Excel spreadsheets, and dietary adherence to the PNE was scored using a validated adherence score sheet (ASS) [34] (unpublished). Adherence to four categories, namely GI, GL, protein and fat, was scored. Scores were allocated based on the adherence to the specific types and amounts of food prescribed in the PNE. A score of ‘1’ was given for adherence to the specific recommendations, and a ‘0’ was scored for non-adherence. Adherence scores (ASs) were calculated for each of the three main meals (breakfast, lunch, and supper) and three in-between meal snacks for adherence to the four macronutrient subgroups, namely GI (low/intermediate), GL, fat and protein. For each macronutrient subgroup, a maximum score of 6 per day and 18 per 3-day record could be achieved. A maximum score of 4 points per meal, 24 per day, and a total AS of 72 for all three days could be achieved. The AS was calculated as the mean of the 3 days per time point.

In addition to the 3-day AS, a main meal adherence classification (MMAC) was used to classify the main meals regarding adherence to the macronutrient subgroups. The ASs for all nine main meals (three main meals per day) in the 3-day food records of all patients collected at time points 1 *(n* = 67), 2 *(n* = 67), and 3 (*n* = 26) were used. In some cases, main meals were skipped, and ASs could not be allocated to these. The numbers of main meals that were recorded over the study period and could be scored added up to a total of 1428. The main meal ASs were classified into the MMAC. The MMAC was used to determine whether adherence to main meals was associated with pre- and postprandial blood glucose levels falling into the targeted ranges of fasting blood glucose levels between 4 and 7 mmol/L and postprandial plasma glucose levels between 5 and 10 mmol/L [33]. The MMAC was classified as follows:

MMAC = 0 if GI = 1 and GL = 1 and both Protein = 1 and Fat = 1      = 1 if GI = 1 and GL = 1 and either Protein or Fat is 1 and the other is 0      = 2 if GI = 0 and GL = 0 and both Protein = 1 and Fat = 1      = 3 if GI = 0 and GL = 0 and either Protein or Fat is 1 and the other is 0, or if all values = 0

An MMAC of ‘0’ indicated adherence to all macronutrient subgroups; ‘1’ indicated that there was adherence to the GI and the GL and either protein or fat; ‘2’ indicated that there was no adherence to the GI or the GL but there was to both protein and fat; and ‘3’ indicated that none of the macronutrient subgroups were adhered to.

The blood glucose control (BGC) measured at main meals was classified according to whether a patient’s pre- and postprandial values fell within the targeted blood glucose ranges [33] as follows:

BGC = 0 if 4 < BGpre < 7 and 5 < BGpost < 10 (start right, end right)    = 1 if BGpre < 4 or BGpre > 7 and 5 < BGpost < 10 (start wrong, end right)    = 2 if 4 < BGpre < 7 and BGpost < 5 or BGpost > 10 (start right, end wrong)    = 3 if BGpre < 4 or BGpre > 7 and BGpost < 5 or BGpost > 10    (start wrong, end wrong)

A BGC of ‘0’ indicated perfect blood glucose control, where the pre- and postprandial values fell within the targeted ranges; ‘1’ indicated that preprandial blood glucose levels were not within the targeted range whereas postprandial values were; ‘2’ indicated that the preprandial values were within the targeted range and postprandial values were not; and ‘3’ indicated that neither pre- nor postprandial blood glucose was within targeted ranges.

### 2.3. The Statistical Analysis

R/RStudio Software (R version 4.4) for statistical analysis was used for the statistical analysis. Descriptive statistics were used to summarise the participants’ characteristics. Simple frequency distributions were used to quantify categorical variables (e.g., gender); measures of the central tendency (mean) and variability (standard deviation) were calculated to describe continuous variables such as age, height, weight, BMI, body fat percentage, and muscle mass.

A one-way repeated measures analysis of variance (ANOVA) was used to determine whether there were statistically significant differences in the anthropometric variables, ASs, mean blood glucose levels, HbA1c values, and dosages of medication at different time points. Tukey’s HSD post hoc test was used to determine at which time points differences occurred. Histograms of the variables, as well as the results of the Shapiro–Wilk test for normality, showed that the variables followed a normal distribution or were close to normality, and therefore the normality assumption for the application of an ANOVA test was not violated. Pearson’s product–moment correlations were used to determine the correlation between the ASs and mean blood glucose and HbA1c values.

Classifications for the MMAC and BGC were used to construct 4 × 4 cross-tabulations. A chi-square test was used to determine the association between the adherence to main meals (as determined by the MMAC) and BGC.

## 3. Results

### 3.1. The Participants’ Characteristics

The 67 patients’ records were included in this study for time points 1 and 2, whereas for time point 3, only 26 patients’ records were included because not all patients (*n* = 41) attended consultation 3 within 90 days ± 2 weeks. Of the 67 patients, 32 (48%) were women. At time point 3, of the 26 remaining patients, 12 (46%) were women. At time points 0, 1, and 2, the mean (±SD) age of the patients was 53.8 ± 11.8 years, and at time point 3, the mean (±SD) age of the patients was 52.4 ± 11.9. Their mean BMI classified them as obese during all three time points (Table 1). Significant differences (*p* < 0.05) in body weight, body fat percentage, and BMI were seen between certain time points. No statistically significant differences in muscle weight were seen (*p* = 0.093).

The medication usage of the patients included a variety of oral medications and different types and regimes of insulin. Their medication usage is described in Table 1. During the study period, 26 (39%) patients reduced their medication, with 10 patients (15%) reducing their oral medication dosages, 15 (22%) reducing their insulin dosages, and 1 reducing both their oral medication and insulin (1%); nine patients decreased their insulin dosages at more than one visit. Insulin dosages were decreased by 1 to 3 units at a time. Four patients (6%) were able to stop injecting insulin at time point 3. Two patients (3%) increased their insulin dosages (between one and three units at a time), and one patient (2%) increased their oral medication during the study period. The reduction in the dosages of oral medication and insulin administration was not statistically significant over the different time points (*p* = 0.615).

### 3.2. The Adherence to Personalised Nutrition Education (PNE)

The total mean AS ranged between 88% and 95%. The mean total AS at time point 1 was 63.7 ± 6.0, indicating a mean adherence of 88% to the PNE, which increased significantly to 66.1 ± 4.9 (92%) at time point 2 and to 95% at time point 3 (Table 2). No statistically significant differences were seen in the adherence scores between genders (*p* = 0.160) or age groups (*p* = 0.660) [results not reported].

The mean ASs for all macronutrient subgroups were 84% and above at all three time points. No significant differences were seen in the ASs for the GI or the GL over the study period. Statistically significant increases in the ASs for protein (*p* = 0.010) and fat (*p* = 0.006) were seen across the time points. The adherence scores for protein increased significantly between time point 1 and time point 2, and the ASs for fat increased significantly between each time point (Table 2).

To evaluate the adherence to main meals, the MMAC was calculated and reported for all time points combined (Table 3). The MMAC indicated that 72% of the patients’ main meals adhered to all of the macronutrient subgroups, while 13% of the patients’ main meals adhered to three of the macronutrient subgroups (GI, GL, and either protein or fat).

### 3.3. Blood Glucose Control

The mean pre- and postprandial 3-day blood glucose levels and HbA1c levels over the study period are summarised in Table 4. The ANOVA testing revealed that the reductions in the 3-day mean blood glucose levels (*p* = 0.013) and HbA1c values (*p* = 0.003) over time were statistically significant. The mean 3-day blood glucose levels were significantly lower at time point 3 compared to those at time point 1 (*p* = 0.016). HbA1c values were not available for all patients or at all time points; therefore, the number of available HbA1c values differed between time points, and it was therefore not possible to determine statistical differences in the HbA1c values between time points. It can, however, be observed that the HbA1c levels showed a trend of decreasing over the study period, with the HbA1c values at time point 1 being above the recommended HbA1c value of 7% [33], while they were within the recommendations at time points 2 and 3.

Table 5 displays how often the blood glucose levels were within the targeted ranges during main meals over the three time points. The majority (60%) of the patients’ blood glucose readings before and 90 min after main meals were within the targeted ranges, with 13% having both pre- and postprandial blood glucose levels not within the targeted ranges.

### 3.4. The Association Between Adherence to PNE and Blood Glucose Control

The correlation between adherence to PNE and blood glucose control is summarised in Table 6. Adherence was correlated with both blood glucose control indicators, mean 3-day blood glucose levels, and HbA1c values. At all three time points, weak negative correlations were seen between ASs and average 3-day blood glucose scores; however, this correlation was only significant for time point 1 (*p* = 0.030) (Table 6).

When comparing HbA1c values with total ASs, a weak negative non-significant correlation was seen at time point 2 (*p* = 0.250), and a weak non-significant positive correlation was seen at time point 3 (*p* = 0.931) (Table 6). Adherence to main meals using the MMAC indicated a statistically significant association between the MMAC scores and BGC (the chi-square test; *p*-value = 0.028).

## 4. Discussion

The current study aimed to determine whether patients with T2DM could adhere to PNE based on GI, GL, and FII principles and whether adherence was associated with blood glucose control.

We found high adherence (>80%) of the T2DM patients to the PNE throughout the study, with their adherence improving over time. A study in Ghana also reported good adherence (71%) of type 2 diabetics to dietary advice [35]. This is in contrast to similar studies in India that reported an adherence level of 54% [36]; three studies in Ethiopia that reported only 34% [37], 38% [38], and 44% [39] adherence; 22% adherence reported in a study from New Zealand [40]; and 16% adherence in a study from Nepal [41]. Demographic factors like gender and age have been shown to influence the adherence to dietary advice for diabetics [42,43]. However, this was not seen in our study.

There are a few factors that may have contributed to our high ASs. Firstly, we followed a PNE approach presented to patients by a registered dietitian. In the *Nutrition Therapy for Adults With Diabetes or Prediabetes* consensus report, evidence suggested that “one-size-fits-all” eating plans are not effective in the management of diabetes but that a personalised approach that considers factors like cultural backgrounds and personal dietary preferences is more effective and acceptable, leading to higher adherence [44]. Research has also suggested that diabetic PNE should preferably be presented by registered dietitians who possess the knowledge and skills to individualise nutrition education based on a patient’s needs, abilities, lifestyle, and resources [45]. Other studies have confirmed that compared to generic diets, PNE has caused better adherence to dietary advice [46].

Secondly, regular follow-up visits (every 30 days ± 2 weeks) were made where misconceptions and questions arising from the initial PNE could be discussed with the patient to improve their understanding. A narrative review on nutrition inventions for T2DM showed that regular education, support, motivation, and addressing individual challenges can improve the adherence to diabetic diets [47]. Other studies have found that factors like irregular diabetes education or a limited number of nutrition education sessions can contribute to non-adherence among patients with diabetes [42]. A study in Ethiopia, where the barriers influencing the dietary adherence of patients with T2DM were examined, found that a lack of dietary education and the inability of the patients to remember dietary recommendations were among the main reasons for poor dietary adherence [48]. A concept analysis of the dietary adherence among adults with T2DM found that after dietary interventions, regular monitoring of patients is essential to identify and prevent relapses to previous dietary habits [7].

Thirdly, the 3-day food records were an important dietary assessment tool, as they assisted the dietitian in identifying areas where patients did not adhere, enabling targeted re-education. Together with the blood glucose records, patients could see how the correct food choices correlated with improved blood glucose control and thereby were encouraged in adherence. This was confirmed by studies indicating that self-monitoring is an individual attribute of dietary adherence among patients with T2DM [7].

Lastly, the follow-up visits were face to face, which have been shown to foster better adherence compared to telephonic follow-up visits [36].

GI [14], GL [16], and FII [27,49] principles have each been proven to be effective in the management of type 2 diabetes. For this study, these principles were combined into the PNE, and the adherence results indicated that sustainable adherence to GI, GL, and FII principles was possible.

In this study, a significant reduction in mean 3-day blood glucose levels and HbA1c values was seen over time, and the majority of the blood glucose readings taken before and after main meals were within the targeted ranges. The reductions in mean blood glucose levels and HbA1c seen in this study were not only statistically significant but also clinically significant, indicating that the results could positively impact diabetes control in the patients. Other studies have also reported improved blood glucose control on diets where GI, GL, or FII principles were implemented. A meta-analysis of dietary approaches’ effect on glycemic control in patients with T2DM found the low-GI diet to be among the top four diets that significantly improved fasting blood glucose and showed promising effects for controlling HbA1c [50]. This was confirmed by two other meta-analyses of controlled dietary trials that found that a low-GI diet improved glycemic control in people with diabetes [51,52]. Another meta-analysis comparing the effect of the GI on HbA1c levels found that choosing low-GI foods (compared to high-GI foods) had a small but clinically useful effect, similar to using pharmacological agents, on the HbA1c levels in patients with diabetes [14]. A systematic review and meta-analysis showed that diets with a low GL decrease insulin levels [53] and can be used in the management of diabetes [54]. A recent scoping review on the application of the food insulin index to the prevention and management of diabetes found that there was an association between an increased dietary insulin index (calculated using the FII) and the development of insulin resistance and T2DM and that the FII was superior to carbohydrate counting in predicting postprandial insulin responses [27]. All this evidence therefore confirms the beneficial impact that the application of GI, GL and FII principles has on blood glucose control in patients with type 2 diabetes.

It must be considered that the GI, GL, and FII principles were presented to patients as part of the PNE, where likes, dislikes, personal circumstances, and beliefs were considered. Studies have shown that personalised nutrition interventions improve blood glucose control more than generic diets [47]. In a cross-over study comparing blood glucose control when participants followed both personalised and non-personalised nutrition programmes, postprandial blood glucose levels were significantly lower and a significant improvement in glycemic control was seen when personalised nutrition was implemented compared to generic diets [55]. A systematic review on randomised controlled trials found that personalised nutrition interventions significantly improved HbA1c and postprandial blood glucose levels compared to non-personalised nutrition interventions [56]. As mentioned before, better adherence to PNE compared to that to generic diets was found [46], and studies have shown that adherence to nutrition education can improve blood glucose control [57,58]. Successful dietary adherence has several positive effects, including improvements in HbA1c levels, lipid profiles, BMI, and blood pressure [59]. In this study, however, weak negative correlations were seen between overall dietary adherence and the average 3-day blood glucose levels, as well as HbA1c levels. However, a significant positive correlation between main meal adherence, as measured by the MMAC score, and BGC was found, indicating that the high adherence to main meals was associated with blood glucose levels that fell within the targeted ranges. Similarly, in a study where low adherence (38%) to diabetic dietary advice was seen, the majority (55%) of the participants failed to achieve the recommended fasting blood glucose target [38].

A significant reduction in weight, BMI, and body fat percentage between certain time points was seen in this study. Studies have indicated that BMI and body fat percentage are associated with fasting blood glucose levels [60]. A study by Stanford et al. found that weight loss can be associated with a decrease in fasting blood glucose levels and HbA1c in patients with diabetes [61]. The participants in the Stanford et al. study lost a mean of 11.7% of their initial body weight, suggesting that to achieve improved blood glucose results, weight loss had to be both statistically and clinically significant [61]. Reductions in weight, BMI, and body fat percentage could have been confounders to blood glucose control in this study, but it is noteworthy that although these changes were statistically significant, the reductions in BMI, weight, and body fat percentages were not clinically significant.

Although medication changes were not an outcome of this study, the results indicate that adherence to PNE has medication-lowering potential given the high proportion of patients for whom medication was lowered. A meta-analysis found that low-GI diets had an effect similar to using pharmacological agents on the HbA1c levels in patients with diabetes [14]. The correlation between implementing low-GI, -GL, and -FII diets and reductions in the dosages of medication could be explored in future research.

A strength of this study is the use of 3-day food records to collect the dietary intake data. Studies have shown that 3-day food records report actual food intake more accurately than other methods, like food frequency questionnaires. Because 3-day food records depend less on memory, fewer errors in food reporting are seen than those for other intake reporting methods [62,63]. Another strength is that the study sample was representative of both genders and most adult age groups.

One limitation of this study is that HbA1c values were not available for all patients in the retrospective data. This limited the statistical analyses of HbA1c. As HbA1c is a long-term blood glucose control indicator, future studies with longer study periods could be conducted. Studies suggest that dietary interventions lasting 4-12 months or longer show significant changes in HbA1c levels compared to those with shorter studies [5]. The study population also only included literate patients with a moderate to high socioeconomic status living in an urban setting and attending a specific private practice. Studies have shown that factors like illiteracy [64] or lower levels of education [35], a lower socioeconomic status [38], low food security [37,64], and living in rural communities [38,65] are factors that negatively affect adherence to diabetes nutrition education. Future studies should be conducted in illiterate and low-socioeconomic-status populations and in a rural setting to investigate the feasibility and adherence for similar PNE.

## 5. Conclusions

This study showed that high adherence to personalised diabetes nutrition education based on GI, GL, and FII principles is possible and that adherence is associated with improved blood glucose control. The findings of this study indicate that healthcare professionals offering nutrition education to people with T2DM should endeavour to provide personalised education to enhance their adherence. More studies on PNE based on GI, GL, and FII principles performed over longer periods are required to confirm the benefits of blood glucose control found in this study.

## Figures and Tables

**Table 1 ijerph-22-00925-t001:** Characteristics of the study sample over different time points (*n* = 67).

Variable	Time Point 0 (Pre-Nutrition Intervention) (*n* = 67)	Time Point 1 (~30 Days Post-Nutrition Intervention) (*n* = 67)	Time Point 2 (~60 Days Post-Nutrition Intervention) (*n* = 67)	Time Point 3 (~90 Days Post-Nutrition Intervention) (*n* = 26)	*p*-Value
Gender	*n* (%)				
Female	32 (48)	32 (48)	32 (48)	12 (46)	-
Male	35 (52)	35 (52)	35 (52)	14 (54)	-
Age (years)	53.8 ± 11.8	53.8 ± 11.8	53.8 ± 11.8	52.4 ± 11.9	-
Height (m)	1.7 ± 0.1	1.7 ± 0.1	1.7 ± 0.1	1.7 ± 0.1	-
Weight (kg)	99.7 ± 21.5 ^abc^	98.5 ± 20.6 ^a^	97.6 ± 20.0 ^bd^	98.0 ± 17.3 ^cd^	<0.001
Body mass index (kg/m^2^)	34.5 ± 6.0 ^abc^	34.1 ± 5.8 ^ad^	33.8 ± 5.6 ^be^	34.5 ± 5.8 ^cde^	<0.001
Body fat percentage	40.8 ± 8.5 ^ab^	40.1 ± 8.8 ^c^	39.2 ± 9.0 ^a^	39.4 ± 9.7 ^bc^	<0.001
Muscle mass percentage	56.6 ± 8.4	57.4 ± 8.6	57.2 ± 10.7	56.55 ± 14.2	0.093
Medication usage	*n* (%)				-
Oral medication	64 (96)	64 (96)	64 (96)	23 (88)	-
Insulin	21 (31)	21 (31)	21 (31)	6 (23)	-

Height, weight, body mass index, body fat percentage, and muscle weight reported as mean ± SD. ^abcde^ Means within a row with a common superscript differ significantly (*p* < 0.05).

**Table 2 ijerph-22-00925-t002:** Total and macronutrient subgroup adherence scores and percentages over time points.

Adherence Score	Time Point 1 (~30 Days Post-Nutrition Intervention) (*n* = 67)	Time Point 2 (~60 Days Post-Nutrition Intervention) (*n* = 67)	Time Point 3 (~90 Days Post-Nutrition Intervention) (*n* = 26)	*p*-Value
Total	63.7 ± 6.0 [88%] ^ab^	66.1 ± 4.9 [92%] ^ac^	68.7 ± 3.7 [95%] ^bc^	0.016
Glycemic Index	16.6 ± 1.5 [92%]	16.2 ± 1.8 [90%]	17.0 ± 1.5 [94%]	0.282
Glycemic Load	15.2 ± 2.8 [84%]	15.5 ± 2.2 [86%]	16.7 ± 1.6 [93%]	0.195
Protein	16.5 ± 1.8 [92%] ^a^	17.3 ± 1.0 [96%] ^a^	17.7 ± 0.7 [98%]	0.010
Fat	15.5 ± 2.1 [86%] ^ab^	17.0 ± 1.4 [94%] ^ac^	17.4 ± 1.1 [97%] ^bc^	0.006

All data reported as mean ± SD [percentage]. Maximum total adherence score = 72; maximum glycemic index/glycemic load/protein/fat adherence score = 18. ^abc^ Means within a row with a common superscript differ significantly (*p* < 0.05)

**Table 3 ijerph-22-00925-t003:** Main meal adherence classification of patients (*n* = 1428).

Classification	Total Meals *n* (%)
Adherence to all macronutrient subgroups	1024 (72)
Adherence to GI and GL and either protein or fat	186 (13)
No adherence to GI or GL, adherence to both protein and fat	163 (11)
No adherence to any macronutrient subgroups	55 (4)

**Table 4 ijerph-22-00925-t004:** Blood glucose control over time points.

	Time Point 1 (~30 Days Post-Nutrition Intervention) (*n* = 67; *n* = 32 *)	Time Point 2 (~60 Days Post-Nutrition Intervention) (*n =* 67; *n* = 17 *)	Time Point 3 (~90 Days Post-Nutrition Intervention) (*n* = 26; *n* = 13 *)	*p*-Value
Mean pre-and postprandial blood glucose (mmol/L)	8.0 ± 2.4 ^a^	7.2 ± 1.5	6.6 ± 0.8 ^a^	0.013
HbA1c (%)	8.4 ± 2.2	6.8 ± 1.0	6.6 ± 1.3	0.003

All data reported as mean ± SD. HbA1c, glycated haemoglobin. * Sample for HbA1c assessment. ^a^ Means differ significantly (*p* < 0.05).

**Table 5 ijerph-22-00925-t005:** Targeted blood glucose control of patients at main meals (*n* = 1428).

Categories	*n* (%)
Blood glucose in the targeted range before and after the meal	859 (60)
Blood glucose not within the targeted range before the meal but in the targeted range after the meal	305 (21)
Blood glucose in the targeted range before the meal, not in the targeted range after the meal	80 (6)
Blood glucose not in the targeted range before or after the meal	184 (13)

**Table 6 ijerph-22-00925-t006:** Pearson’s correlation of blood glucose control indicators with total adherence scores at different time points.

	Correlation (r): Blood Glucose vs. Total Adherence Score	*p*-Value	Correlation (r) HbA1c vs. Total Adherence Score	*p*-Value
Time Point 1 (*n* = 67; *n* = 32 *)	−0.265	0.030	-	-
Time Point 2 (*n* = 67; *n* = 17 *)	−0.188	0.127	−0.295	0.250
Time Point 3 (*n* = 26; *n* = 13 *)	−0.032	0.880	0.028	0.931

* Sample for HbA1c assessment; *p* ≤ 0.05 indicates statistical significance.

## Data Availability

The raw data supporting the conclusions of this article will be made available by the authors on request.

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
