# Peer review of "Adherence to Personalised Nutrition Education Based on Glycemic and Food Insulin Index Principles and Their Association with Blood Glucose Control in Individuals with Type 2 Diabetes Mellitus"

_ijerph, 2025, doi:10.3390/ijerph22060925_

Round 1

Reviewer 1 Report

Comments and Suggestions for Authors
  • Though blood glucose was measured, ethical clearance is needed to furnish properly.
  • In line 135 what is meant by t 2?
  • Is the patient being under any medication including T2DM? If yes, then is this result acceptable that PNE helps to reduce glucose? Please clarify
  • Need to add implication and future prospects of this study at conclusion?

Reviewer 2 Report

Comments and Suggestions for Authors

Thank you for the submission of the manuscript Adherence to personalised nutrition education based on glycaemic index and the food insulin index principles and association with blood glucose control in type 2 diabetics.

Suggested title amendment

Adherence to personalised nutrition education based on glycaemic index and the food insulin index principles and association with blood glucose control in individuals with type 2 diabetes mellitus.

Consistency required with abbreviation T2DM (type 2 diabetes mellitus) or T2D (type 2 diabetes). Line 34, 43 and 72 where it says type 2 diabetes please change to the preferred abbreviation.

Line 92 – 93 it is mentioned that the Tanita Body Composition Monitor BC-731 was utilised to measure body fat and muscle mass percentage, how accurate is this? As DEXA is the gold standard for monitoring body composition.

Line 93 it has been observed in table 1 that oral medication and insulin is included in the table. What is the medication that is utilised for oral medication? For example, are all the participants taking metformin? Or is there a variation, further, is insulin one type, or does the insulin vary with a number of different types included, long acting, short acting etc?

Line 101 what is the brand of the glucometer?

Line 119 – 120, why is there a variation the in the fat and protein contents for males and females for the main meals but not the snacks.

Line 199 Table 1

Sex male and female n should be added into the table as well as age ± SD across the time points, it would be interesting to see the reduction in the change for the participants for time point 3 and the distribution in the table as well as the text.

Why is the body fat in percentage and the muscle mass in kilograms? They should both be consistent.

It is suggested that where it Time Point 1, Time Point 2 and Time Point 3 is at the beginning of the table also includes Baseline/ Pre nutritional intervention (Time Point 1), ~30 days post nutritional intervention (Time Point 2), ~60 days post nutritional intervention (Time Point 2) and ~90 days post nutritional intervention (Time Point 3).  This should be applied to all of the tables in the results section to make it clearer to the reader.

Fibre also influences glycaemic index, why was this not controlled for or mentioned in the study and only the macronutrients?

Table 4 shows the blood glucose controlled over time points, there is a markedly lower amount of participants that had their blood glucose measured and HBA1c measured, therefore, what is the difference in the characteristics compared to Table 1 when there is an n=67 for the first 3 time points and then n=26 for time point 4 compared to an n of 32, 17 and 13 and further why was there no base line reading particularly for HBA1c?  BMI and body composition may vary given that the sample size is a lot smaller for each of the groups and may also be influencing this. It would be worth analysing this data to compare when highlighting the blood glucose concentration data.

If data was analysed from medical records, wouldn’t it be ethical to obtain consent from the participants for the data to be utilised in the study even if de-identified?

The discussion section of the journal article is quite short and could be elloborated on.

Reviewer 3 Report

Comments and Suggestions for Authors

The authors have put good efforts for the research works. The content is fine. Still i have some queries and comments regarding the work.

First of i have concern regarding the ethical issue regarding the consent from the participants. Even though this is a retrospective cohort study, still the people who did intervention with the patients should have taken written consent from the patients. The authors could have mentioned about it. 

It is necessary to elaborate about the "3-day food record". Is it done every 3 day or at the time of visit to the center? Is there any gap between the 3-day record?

In the table for there is no information about the glucose level whether it is average of per-prandial or post-prandial or combination of both.

The sample size of HbA1c  is lesser than the data of blood glucose level. Please clarify about it.

Page 3, line 129-129: There is error in Mathematical calculation. If 4 pints are given per meal then 3 meals would get 12 point/day. Please check it.

It would be nice to illustrate the actual meaning regarding the figures of GI, GL, Protein, Fat as 1 or 0. What activity of patient would get such point 0 or 1. 

Author Response

Please see attchment

Round 2

Reviewer 1 Report

Comments and Suggestions for Authors

Comments:

  • In methodology duration PNE was provided but any duration/visit no information was not provided.
  • In line 111 “All 18 blood glucose readings “, 18 means what here? please clear.
  • In line “A total of 1428 main meal”, please mention how have you get 1428 no?
  • Table 6 need to be written more understandable like p -value for which parameter need to be more understandable.
  • P value significant point needs to be written in table 6.
  • Reference no 34 is an unpublished self-cited reference. Please replace it.
Comments on the Quality of English Language

Some grammatical error found , please recheck it. 

Reviewer 2 Report

Comments and Suggestions for Authors

Thank you to the authors for their work and taking the time to make the amendments and confirming that there was in fact ethical approval for this study. Did the participants provide informed consent albeit the study being retrospective? 

The reference [34] where the authors have cited their own paper should also indicated in the manuscript itself that this is unpublished, or this appears to be a reference.

Great to see Brand-Miller referenced in this paper, as this work is quite relevant to the area of glycaemic index work.

Reviewer 3 Report

Comments and Suggestions for Authors

Everything is fine.

Author Response

Thank you for your review